# Overexpression of *LAR1* Suppresses Anthocyanin Biosynthesis by Enhancing Catechin Competition Leading to Promotion of Proanthocyanidin Pathway in Spine Grape (*Vitis davidii*) Cells

**DOI:** 10.3390/ijms252212087

**Published:** 2024-11-11

**Authors:** Junxuan Lin, Gongti Lai, Aolin Guo, Liyuan He, Fangxue Yang, Yuji Huang, Jianmei Che, Chengchun Lai

**Affiliations:** 1Institute of Food Science and Technology, Fujian Academy of Agricultural Sciences, Fuzhou 350003, China; 17857696976@163.com (J.L.); laigongti@faas.cn (G.L.); guoaolin3425125034@163.com (A.G.); 18396531571@163.com (L.H.); 13648356144@163.com (F.Y.); 2College of Horticulture, Fujian Agriculture and Forestry University, Fuzhou 350002, China; 3Fujian Key Laboratory of Agricultural Product (Food) Processing, Fuzhou 350003, China; 4Institute of Resources, Environment and Soil Fertilizer, Fujian Academy of Agricultural Sciences, Fuzhou 350003, China; chejianmei@faas.cn

**Keywords:** spine grape (*Vitis davidii*), proanthocyanidins, leucoanthocyanin reductase (LAR), overexpression, light quality

## Abstract

Proanthocyanidins (PAs) are a class of polyphenolic compounds recognized for their potent antioxidant, anti-cancer, anti-inflammatory, and cardioprotective properties. However, the production of PAs from natural sources is often limited by high costs, resource wastage, and environmental damage. In this study, we investigated the overexpression of *VdLAR1*, along with phenotypic observation, metabolite determination, light quality treatment, and RT-qPCR analysis, in spine grape cells. The results demonstrated a significant increase in the contents of proanthocyanidins and flavonoids in p*VdLAR1*-overexpressing transgenic cell lines, while anthocyanin levels showed a decreasing trend. Furthermore, the treatment with white and blue light on the T5 cell line resulted in enhanced accumulation of proanthocyanidins, catechins, and flavonoids, whereas anthocyanins and epicatechins exhibited a declining pattern. Thus, short-wavelength light promoted the accumulation of metabolites, with the proanthocyanidin content in the T5 transformed cell line reaching 2512.0 μg/g (FW) during blue light incubation. RT-qPCR analysis revealed that the key genes involved in the biosynthesis of proanthocyanidin and anthocyanin were upregulated in the transgenic spine grape cell lines, with *VdLAR1* expression increasing by several hundredfold, far surpassing the expression levels of *LDOX* and *ANR*. The *VdLAR1* overexpression markedly improved substrate competitiveness within the metabolic pathway, promoting catechin biosynthesis while inhibiting the production of epicatechins and anthocyanins. This finding provides compelling evidence that *LAR1* is a crucial gene for catechin biosynthesis. This research establishes both theoretical and practical foundations for the regulation and development of natural proanthocyanidins, addressing issues related to high costs, safety concerns, resource wastage, and environmental damage associated with their production.

## 1. Introduction

Proanthocyandins (PAs) are a type of polyphenolic compound present in plants, and are produced through the polymerization of flavan-3-ol monomers [1,2]. PAs and their monomers play crucial roles in human health, exhibiting immunomodulatory and anti-cancer activities derived from their high antioxidant and free radical scavenging capacity [3,4]. As a result, they are crucial in mechanisms associated with anti-inflammation processes, cardioprotection, thrombosis prevention, and protection against ultraviolet rays [5,6]. Therefore, proanthocyanidins have a wide range of applications and high demand in the food, pharmaceutical, and cosmetic industries. The primary source of natural proanthocyanidins is from plants, such as grape, cranberry, wolfberry, and pomegranate [3,7]. Among these, the extraction of proanthocyanidins from grape seeds represents a crucial approach for acquiring these compounds and possesses significant market potential [8]. The skins and seeds of spine grape (*Vitis davidii* Foëx.) are abundant in a variety of bioactive compounds, offering essential raw materials for the extraction and application of proanthocyanidins [9]. Additionally, their significant genetic resources merit further exploration and utilization. The biosynthesis of proanthocyanidin utilizes the same upstream phenylpropanoid pathway as well as the fundamental flavonoid pathway associated with anthocyanins. The specific pathway for proanthocyanidin diverges into two branches, each involving distinct enzymes: leucoanthocyanin reductase (LAR) and anthocyanidin reductase (ANR) [10]. LAR is a key enzyme in the biosynthesis of proanthocyanidins, catalyzing the generation of flavan-3-ol monomers and further polymerization to form proanthocyanidin polymers [11,12]. To date, LAR genes have been isolated from various plants, including apple, tea, and cacao, among others [13,14,15,16]. The ectopic expression of the apple *MdLAR1* gene in tobacco was significantly correlated with the contents of catechin and resulted in a loss of anthocyanin in flowers [17]. Overexpression of *CsLAR* in tobacco led to an increase in catechins while markedly decreasing anthocyanin accumulation [18]. Transgenic tobacco overexpressing *TcLAR* had decreased amounts of anthocyanidins and increased PAs [19]. Moreover, the biosynthesis of proanthocyanidin is regulated not only by its own structural genes and transcription factors, but also by environmental factors during its metabolic processes. Among these factors, light quality plays a significant role in affecting the pigment content and composition in plants [20,21,22]. Red light (RL, 620–700 nm) and blue light (BL, 400–500 nm) in the visible region of the radiation spectrum are the most effective for photosynthesis and therefore important for plant growth [23]. Blue light has been found to have a significant effect on grape flavonoid biosynthesis, especially flavonol and anthocyanin biosynthesis [24,25,26]. However, studies on the light response of *LAR* and the light quality regulation of proanthocyanidin biosynthesis in spine grape are limited and warrant further investigation.

Natural sources of proanthocyanidins are primarily plant tissues such as grape seeds. However, this extraction process faces several challenges, including limited resources, low concentration, complex methodologies, environmental degradation, and high cost. Consequently, there is a pressing need to explore entirely new alternative or complementary solutions to satisfy the increasing demand for proanthocyanidins in the food, pharmaceutical, and cosmetic industries. To address this issue, we propose a strategy for the production of natural proanthocyanidins using plant cell factories through cell culture and gene engineering. In this study, we cloned the *VdLAR1* gene from spine grape, and performed genetic transformation to investigate the effect of the *VdLAR1* gene on the accumulation of proanthocyanidins, anthocyanins, and flavonoids. Furthermore, transformed positive calli that exhibited high proanthocyanidin production were screened and subjected to various light quality treatments. The levels of product accumulation and the expression profiles of crucial genes in the proanthocyanidin biosynthesis pathway were subsequently assessed. The results established a foundational basis for the production of proanthocyanidins through plant cell factories and clarified the regulatory mechanism of the *VdLAR1* gene in proanthocyanidin biosynthesis in spine grape.

## 2. Results 

### 2.1. Gene Cloning and Sequence Analysis of VdLAR1

Sequence analysis revealed that *VdLAR1* (GenBank accession number: PQ185972.1) possesses a full-length open reading frame (ORF) of 1041 bp, encoding a polypeptide consisting of 346 amino acids. The amino acid composition and physicochemical properties of the VdLAR1 protein were analyzed using ExPASy-Protparam. The results indicated that the molecular weight of VdLAR1 protein was 38.03 kDa, the isoelectric point (pI) was approximately 5.61, and the instability index was 41.07, suggesting that VdLAR1 was an unstable protein. Furthermore, the grand average of hydropathicity (GRAVY) of VdLAR1 was −0.077, indicating that the protein is hydrophobic (Figure 1A). Predictions made using SignalP-4.1 showed that VdLAR1 had no signaling peptide (Figure 1B). The SOPMA secondary structure prediction revealed that the VdLAR1 protein is primarily composed of α-helix (35.84%), β-fold (6.36%), irregularly coiled (41.04%), and extended chain (16.76%) structures (Figure 1C). Additionally, the analysis of conserved domains of the VdLAR1 protein using the NCBI-CDD revealed the presence of phenylcoumaran benzylic ether reductase (PCBER), which belongs to the short-chain dehydrogenase/reductase (SDR) superfamily (Figure 1D). Phylogenetic tree analysis demonstrated that the VdLAR1 sequence exhibited the highest homology with VvLAR1 (98.75%) and VbLAR1 (98.46%) (Figure 1E). 

### 2.2. Subcellular Localization and Overexpression of VdLAR1

In order to further verify its subcellular localization characteristics and gene function, a recombinant expression vector, *pBI121-VdLAR1-EGFP*, was constructed by fusing the target gene with the expression vector (Figure 2A). Transient transformation in protoplasts demonstrated the presence of GFP fluorescent signals in the cytoplasm, suggesting that the VdLAR1 protein was localized in the cytoplasm and may perform its function in this organelle (Figure 2B). The recombinant plasmid was introduced into spine grape cells via *Agrobacterium*-mediated transformation, resulting in the initial identification of five resistant transgenic cell lines under kanamycin (Kan) selection. These cell lines showed bright green fluorescent protein signals visible to the naked eye under fluorescent light. The DNA-PCR amplification confirmed that the electrophoretic bands of all the transformed cell lines matched the expected results, while the WT exhibited no electrophoretic band due to the absence of vector sequences harboring the targeted *LAR1* gene (Figure 2C). Consequently, five transformed positive calli were successfully obtained through fluorescence observation and DNA-PCR assay screening (Figure 2D).

### 2.3. Overexpression of VdLAR1 Promotes Proanthocyanidin Accumulation in Spine Grape Cell Lines

The *VdLAR1*-transformed cells and WT cells exhibited no significant differences in size, shape, or texture for different culture durations (Figure 3A). All cell lines demonstrated pigment accumulation with prolonged culture durations. However, the transformed cells displayed a lighter surface color, indicating an inhibition of anthocyanin accumulation. Metabolite determination revealed a significantly higher proanthocyanidin content in transgenic cell lines compared to WT, except for T4 at the late culture stage (35 d) (Figure 3B). Furthermore, as the duration of cultivation was extended, a noticeable increase in accumulation was detected. Among these transformed cell lines, T5 demonstrated the highest accumulation of proanthocyanidin, which reached the peak concentration of 1960.89 μg/g (FW) at the late culture stage (35 d), an increase of 1.7-fold. The flavonoid levels showed a significant enhancement in the transformed cell lines, with a maximum increase of 3.54-fold at the late culture stage (35 d) (Figure 3B). Nevertheless, as the duration of cultivation extended, the accumulation of products exhibited only a gradual rise. In contrast, the accumulation pattern of anthocyanin differed from that of proanthocyanins and flavonoids, exhibiting decreased levels in the transformed cell lines compared to WT (Figure 3B). Anthocyanins tend to accumulate over time, reaching a peak concentration of 93.27 μg/g (FW) in the WT at the late culture stage (35 d). The T5 cell line also demonstrated relatively high levels at 90.52 μg/g (FW), whereas the T3 cell line exhibited only 40.9% of the WT content, measuring 54.09 μg/g (FW). In summary, overexpression of *VdLAR1* significantly enhanced the biosynthesis of proanthocyanidin and flavonoid while inhibiting the accumulation of anthocyanin in spine grape cell lines. Among these, the T5 transformed cell line exhibited the highest level of proanthocyanidin accumulation, thus providing a material basis for studying the regulation of proanthocyanidin biosynthesis.

### 2.4. Effects of Light Quality on Proanthocyanidin Accumulation in VdLAR1 Overexpression Cell Lines

Light quality plays an important role in plant growth and metabolite accumulation. The four light qualities produced varying effects on growth in WT and the T5 transgenic cell line (Figure 4A). The T5 cell line showed a distinct deep red color under white and blue light treatments, while it displayed a light pink color when exposed to red light. In contrast, the T5 cell line cultured in darkness exhibited no color accumulation. The phenotypes of the T5 cell line and WT were similar across different culture stages, demonstrating comparable responses to varying light quality. Therefore, light is essential for morphogenesis in spine grape cell lines, with blue light, characterized by short wavelengths, significantly enhancing pigment accumulation compared to red light. *VdLAR1* overexpression markedly promoted the accumulation of proanthocyanidin, which was significantly higher under white and blue light treatments (Figure 4B), and the accumulation exhibited a pattern of an increase followed by a decrease, peaking at the mid-stage (25 d) with values of 2144.59 μg/g (FW) and 2512.00 μg/g (FW), respectively. However, the proanthocyanidin accumulation remained below 1000 μg/g (FW) during various periods under red light and darkness. Flavonoid assays revealed that *VdLAR1* overexpression enhanced flavonoid content, exhibiting accumulation trends similar to those of proanthocyanidins. These levels were significantly higher under white and blue light treatments compared to red light and darkness. The flavonoid content of the T5 cell line initially increased, peaked at 2257.25 μg/g (FW) and 2441.57 μg/g (FW) at the mid-stage (25 d) under white and blue light, and subsequently decreased. Anthocyanin accumulation was inhibited by the overexpression of *VdLAR1* (Figure 4B). The WT and T5 cell lines subjected to white light treatment showed continuous accumulation of anthocyanin, reaching a maximum at the late culture stage (35 d). However, anthocyanins under blue light treatment exhibited an increasing trend followed by a decrease, peaking at 75.34 μg/g (FW) at the mid-culture stage (25 d), which is only 66.67% of the WT level. Meanwhile, the accumulation of anthocyanin in both cell lines under red light and darkness remained below 20 μg/g (FW) at various time points. These results indicate that proanthocyanidins, anthocyanins, and flavonoids were induced by white and blue light, suggesting that short-wavelength light promotes the biosynthesis and accumulation of these target metabolites. Additionally, *VdLAR1* significantly increased the content of proanthocyanidin and flavonoid in the transformed cell lines while inhibiting anthocyanin biosynthesis.

Catechins and epicatechins were quantified to further investigate the effect of light quality on the biosynthesis of proanthocyanidin monomers (Figure 4C). *VdLAR1* overexpression significantly enhanced catechin accumulation but inhibited that of epicatechin under white and blue light. In the T5 cell line treated with white light, catechin content initially increased, followed by a decrease, reaching a maximum of 3.64 ng/g (FW) during the mid-culture stage (25 d). Catechin presented a gradual enhancement under blue light with extension of the culture duration, reaching a peak of 3.28 ng/g (FW) at the late culture stage (35 d). However, the epicatechin assay revealed that the epicatechin content in the T5 cell line was significantly lower than that in WT, showing a decreasing trend under white light treatment. Conversely, under blue light treatment, the epicatechin content displayed a pattern of an initial decrease followed by an increase. These results suggest that overexpression of *VdLAR1* promotes catechin accumulation while inhibiting epicatechin biosynthesis, and that different light qualities exert distinct effects on the accumulation patterns of these two proanthocyanidin monomers.

### 2.5. Effects of Different Light Qualities on the Expression of Crucial Genes for Proanthocyanidin Biosynthesis

The *DFR*, *LDOX*, *LAR1*, *LAR2*, *ANR*, and *UFGT* genes are essential structural genes involved in the biosynthesis of proanthocyanins and flavonoids. The expression levels of these genes were detected to further analyze the molecular mechanism underlying proanthocyanidin biosynthesis by *VdLAR1* in response to different light qualities. RT-qPCR results revealed that the expression levels of the *VdDFR*, *VdLDOX*, *VdLAR1*, *VdLAR2*, *VdANR*, and *VdUFGT* genes were significantly higher in the T5 cell line compared to WT. However, exceptions were observed for *VdLAR2* under white light, blue light, and darkness at the late culture stage (35 d), as well as for *VdANR* under darkness at this stage and for *VdUFGT* under white light at the early culture stage (15 d) (Figure 5A). Notably, there was a several-hundred-fold increase in the expression of *VdLAR1* under distinct light conditions, with the expression levels markedly upregulated and peaking at 573-fold under blue light treatment at the late culture stage (35 d). The expression of *VdLAR1* increased and then decreased under white light, while it consistently accumulated under blue light, aligning closely with the trend observed in catechin accumulation. The expression of the *VdDFR* gene in the T5 cell line was relatively similar to that of WT during the mid-culture stage (25 d). However, it showed a slight increase in the early and late stages (15 d and 25 d), with the highest expression observed at the late culture stage (35 d) under the blue light treatment, reaching a 3.3-fold increase. *VdLAR1* overexpression also led to the upregulation of *VdLDOX* under both white and blue light, achieving 4.35 and 12.37 times that of WT, respectively. In contrast, *VdLAR2* presented a significant downregulation at the late culture stage (35 d) under white light, maintained a low expression level under blue and red light, and displayed high expression at the mid-culture stage (25 d) in darkness. The expression levels of *VdANR* and *VdUFGT* increased under both white and blue light, while remaining at low expression levels under red light and in darkness. *VdANR* exhibited an increased expression pattern at all three cultivation stages (15 d, 25 d, and 35 d) compared to WT. Meanwhile, *VdUFGT* showed a similar expression pattern, except at the early cultivation stage (15 d), where it displayed a low level of expression under white light. The results indicate that short-wavelength light quality has a significant promoting effect on the biosynthesis of proanthocyanin genes. *VdLAR1* enhances the competitive metabolism between the *LAR* (catechin pathway) and *LDOX-ANR/UFGT* pathways (epicatechin and anthocyanin) (Figure 5B). However, the catechin pathway dominated the competition for substrates, resulting in substantial enrichment of target metabolites due to a hundred-fold increase in *VdLAR1* expression, which subsequently inhibited the production of epicatechin and anthocyanin.

## 3. Discussion

### 3.1. Sequence and Structural Characterization of VdLAR1

Spine grape berries are purple-black in color and are rich in various biologically active components, including proanthocyanidins, which not only provide unique nutritional value but also serve as high-quality raw materials for foods, pharmaceuticals, and other fields, thus possessing significant research value [27]. The LAR gene is a structural gene that regulates the biosynthesis of proanthocyanidin. To date, the LAR gene has been cloned from various plants, including *Vitis vinifera*, *Malus* x *domestica*, and *Populus trichocarpa* [28]. Two functionally similar *LAR* gene members, *VvLAR1* and *VvLAR2*, were successfully cloned from *Vitis vinifera*; they not only promote the biosynthesis of flavan-3-ol monomers but also play an important role in regulating proanthocyanidin accumulation [29]. Two *LAR* genes were cloned from *Malus* x *domestica*, and MdLAR protein was obtained through heterologous expression in yeast, and the results indicated that both *MdLAR1* and *MdLAR2* could enhance the production of catechins and epicatechins [30]. Analysis of the *PtrLAR3* gene, cloned from poplar, revealed that its expression level corresponded with the accumulation pattern of proanthocyanidin [31]. In addition, previous studies reported that the active site of the LAR protein in grapes binds an NADPH, providing a theoretical basis for elucidating the catalytic mechanism of LAR [32]. However, the progress of research on the *VdLAR1* gene in spine grape remains unclear. In this study, we successfully cloned the leucoanthocyanidin reductase gene (*VdLAR1*) containing a complete open reading frame (ORF). The bioinformatic analyses indicated that the coding sequence (CDS) of the *VdLAR1* gene was 1041 bp long and encoded 346 amino acids. VdLAR1 exhibited high sequence homology with LAR proteins from other species and contained a typical structural domain of phenylcoumaran benzylic ether reductase (PCBER), suggesting its dehydrogenase characteristics. Phylogenetic analysis revealed that the VdLAR1 protein was clustered with VvLAR1 and VbLAR1 proteins, indicating that they belong to the same protein class and perform similar functions. Regarding the subcellular localization of VdLAR1, all three prediction tools (Cell-PLoc 2.0, CELLO, and WoLF PSORT) indicated that VdLAR1 was localized in the cytoplasm. This finding was corroborated by the results of the transient transformation into protoplasts conducted in this study. Moreover, the integration of bioinformatics analysis and the examination of LAR subcellular localization across different species further confirmed the presence of LAR proteins in the cytoplasm [33,34,35]. These results indicate that VdLAR1 is a typical cytoplasmic enzyme that plays an important role in NADP-dependent reduction reactions.

### 3.2. VdLAR1 Promotes Proanthocyanidin Biosynthesis and Inhibits Competitive Anthocyanin Accumulation

Proanthocyanidins serve not only as a plant defense against biotic and abiotic stresses, such as microbial pathogens, insects, and ultraviolet light, but also exhibit excellent antioxidant properties, free radical scavenging abilities, and other physicochemical activities. These compounds play an important role in human medicine and the general health industry, with recent studies demonstrating their high safety and efficacy in preventing and assisting in the treatment of cardiac diseases [36,37,38]. The biosynthesis of proanthocyanidin precursors shares upstream phenylpropanoid and core flavonoid pathways with anthocyanins. Dihydroflavonols are converted into leucoanthocyanidin under the catalysis of DFR, while both *LAR* and *LDOX* utilize these substrates to synthesize catechins and anthocyanidins, respectively. Furthermore, anthocyanins are subsequently converted into epicatechins by the action of *ANR*, and anthocyanins are also further modified by *UFGT* [19,39]. Therefore, a competitive relationship exists between proanthocyanidin and anthocyanin biosynthesis, as the LAR and LDOX enzymes compete for the same substrate—leucoanthocyanidins. The activity of LAR influences catechin biosynthesis, while LDOX regulates the production of epicatechins and anthocyanins [40,41]. Gene overexpression has been widely employed in functional studies of plants due to its capacity to generate corresponding traits. For example, the overexpression of *LAR* demonstrated distinct patterns of proanthocyanidin and anthocyanin accumulation [42,43,44]. The overexpression of *MeLAR* in cassava (*Manihot esculenta*) resulted in a decrease in anthocyanin content and an increase in proanthocyanidin levels [45]. Similarly, the overexpression of *TcLAR* in tobacco led to reduced anthocyanin accumulation while promoting the production of epicatechin and catechin monomers [19]. Additionally, *MrLAR* in tobacco also caused a lower level of anthocyanin and enhanced accumulation of proanthocyanidin [46]. 

To investigate the potential function of the *VdLAR1* gene in spine grape, this study successfully overexpressed the *VdLAR1* gene in spine grape cell lines. The results revealed that the cell line with the highest proanthocyanidin content among the five transformed cell lines could reach a 5.5-fold increase compared to the WT. Additionally, the cell line with the highest flavonoid accumulation reached 3.5-fold that of the WT; however, all transformed cell lines showed a downregulation of anthocyanin accumulation. The content of catechin monomers significantly increased in the transformed cell line, while epicatechin displayed the opposite accumulation pattern. Gene expression analyses revealed that crucial genes involved in proanthocyanidin and anthocyanidin biosynthesis were upregulated. Notably, *VdLAR1* not only enhanced the associated metabolic pathway but also promoted the competing metabolic pathway. However, the expression of *VdLAR1* was upregulated several-hundred-fold, far exceeding the expression levels of *LDOX* and *ANR*. This indicates strong substrate competitiveness within the metabolic pathway, significantly enhancing catechin biosynthesis while inhibiting the production of epicatechins and anthocyanins. In summary, the overexpression of *VdLAR1* suppresses anthocyanin biosynthesis by increasing catechin competition, thereby promoting the proanthocyanidin pathway in *V. davidii.* This result provides robust evidence that *LAR1* is a crucial gene for catechin biosynthesis. As a result, the findings demonstrated that the overexpression of *VdLAR1* significantly enhanced the levels of both monomeric anthocyanins and total proanthocyanidins in the cell lines. Previous studies have indicated that proanthocyanidins are abundant in spine grape skin and seeds, with concentrations of approximately 50,000 μg/g (DW) and 10,000 μg/g (DW), respectively [47,48]. Notably, the proanthocyanidin content in the overexpressing *VdLAR1* cell line can reach up to 2000 μg/g (FW), and due to a water content exceeding 95%, the dry weight concentration of proanthocyanidins in these samples surpasses 40,000 μg/g (DW), which is significantly higher than that found in grape seeds and comparable to that present in skins [49]. Furthermore, the high costs associated with the removal of skins and seeds from spine grapes, coupled with seasonal growth limitations, adversely affect the production efficiency of proanthocyanidins derived from these sources [50]. In contrast, callus cultures remain unaffected by external environmental conditions, allowing for year-round continuous production [51]. Therefore, transforming calli for efficient proanthocyanidin production through overexpression presents promising application prospects.

### 3.3. Short-Wavelength Light Promotes Proanthocyanidin Accumulation by Upregulating the Expression of Crucial Genes

Light quality directly influences plant cell growth and the synthesis and accumulation of secondary metabolites during the establishment of efficient plant regeneration systems. Additionally, visible light can induce proanthocyanidin biosynthesis and affect its composition [25,52,53]. Previous studies have demonstrated that blue light significantly increases the content of polyphenolic metabolites, such as proanthocyanidins, while red light exhibits the opposite effect [54]. Short-wavelength light, including blue light and violet light, promotes the accumulation of proanthocyanidin in *V. davidii* cells [9,55]. Meanwhile, varying wavelengths of light affect flavonoid metabolism in grapevine and influence the activities of crucial enzymes involved in flavonoid synthesis [56,57,58]. The expression levels of *LDOX* and *DFR* in *Vaccinium corymbosum* were significantly elevated under blue light, which facilitated the biosynthesis of anthocyanin [59]. Prolonged exposure to blue and green light resulted in an increase in *PAL* gene expression in *Cyclocarya paliurus* and enhanced flavonoid accumulation [60]. Therefore, short-wavelength light, such as blue light, can enhance the accumulation of metabolites by inducing the expression of crucial structural genes. Furthermore, the two *LAR* genes in young berries of Cabernet Sauvignon grape exhibited different responses to light; specifically, *VvLAR1* was light-induced while *VvLAR2* was light-insensitive. These findings provide a foundation for investigating the expression of crucial genes involved in proanthocyanidin biosynthesis and the regulation of products induced by light quality [25,61]. Our study demonstrated that the contents of proanthocyanidins, anthocyanins, and flavonoids in the transformed cell lines under the four light quality treatments were significantly higher than those in the WT. Furthermore, the contents of these three metabolites in the transformed cell lines treated with white light and blue light were more than 1.5-fold greater than those observed under red light and dark treatments. Gene expression analysis was consistent with the aforementioned product quantification. The expression levels of crucial genes involved in proanthocyanidin and anthocyanin biosynthesis, including *VdDFR*, *VdLDOX*, *VdLAR1*, *VdLAR2*, *VdANR*, and *VdUFGT,* were significantly upregulated in the transformed cell lines under white and blue light treatments. Notably, the expression level of the *VdLAR1* gene exhibited a remarkable increase, reaching 573.7-fold of that of the WT. We observed that both white and blue light induced catechin accumulation while inhibiting epicatechin production in the transformed cell lines. These findings align with previous studies indicating the *LAR* gene is associated with catechin biosynthesis and plays a role in the inhibition of epicatechin biosynthesis [62,63,64]. In conclusion, short-wavelength light quality significantly enhances gene expression, subsequently leading to increased metabolite accumulation.

## 4. Materials and Methods

### 4.1. Plant Material and Treatments

The spine grape (*Vitis davidii* Foëx.) cell line “DLR”, previously reported in [65], was induced from immature embryo, and is capable of synthesizing proanthocyanins and anthocyanins exhibiting a red phenotype particularly in the mature stage. The “DLRLJ” cell line, derived from “DLR”, was preserved in our laboratory and utilized in this study. Spine grape cells were cultured on solid MS medium (Murashige and Skoog medium) supplemented with 1.0 mg/L 2,4-dichlorophenoxyacetic acid (2,4-D), 30 g/L sucrose, and 6g/L agar (pH5.8). The subculture period of spine grape cells was 20 days and the cells were grown under a 12/12 h light/dark cycle at 25 °C.

Four types of light quality were employed: white light (mix light), blue light (400–500 nm), red light (620–700 nm) and darkness, with different technical parameters for light quality as previously described [55]. The light source was provided through light-emitting diode (LED) lamps (Leesa LED lamp, Zhongshan, China). Spine grape cells were exposed to different light qualities after 10 days of incubation under white light. Cellular materials were collected at the three culture stages of 15, 25, and 35 days. The cells were snap-frozen in liquid nitrogen and stored at −80 °C.

### 4.2. RNA Extraction and cDNA Synthesis

Total RNA was extracted from the “DLRLJ” cell line using the RNAprep Pure Plant Kit (TIANGEN, Beijing, China). The concentration and quality of the RNA were evaluated using the CLARIOstar Microplate Reader (BMG LABTECH, Ortenberg, Germany) and via agarose gel electrophoresis. For gene cloning, cDNA was synthesized using the EasyScript^®^ One-Step gDNA Removal and cDNA Synthesis SuperMix Kit (TransGen Biotech, Beijing, China). Additionally, cDNA for the RT-qPCR assay was generated using the PrimeScriptTM RT Reagent Kit with gDNA Eraser Kit (TaKaRa, Dalian, China).

### 4.3. Gene Cloning and Sequence Analysis

Utilizing the grapevine genome sequence (Genoscope 12X v2.1) referenced in March 2023, *VdLAR1*-specific primers were designed using Primer Premier 5.0 to amplify the full-length cDNA and are listed in Appendix A. The amplification product was cloned into the pLB vector (TIANGEN, Beijing, China) and confirmed by PCR and sequencing (TsingKe, Xiamen, China). An analysis of the basic physicochemical properties, including hydrophilicity/hydrophobicity, signal peptide, and transmembrane structure, as well as prediction for secondary and tertiary structure of the VdLAR1 protein, was carried out using ExPASy (https://web.expasy.org/protparam/, accessed on 15 April 2024), ProtScale (https://web.expasy.org/protscale/, accessed on 15 April 2024), SignalP 4.0 (https://npsa-pbil.ibcp.fr/cgi-bin/npsa_automat.pl?page = npsa_sopma.html, accessed on 15 April 2024), TMHMM-2.0 (https://services.healthtech.dtu.dk/services/TMHMM-2.0/, accessed on 15 April 2024), SOPMA (https://npsa.lyon.inserm.fr/cgi-bin/npsa_automat.pl?page=/NPSA/npsa_sopma.html, accessed on 15 April 2024), and Swiss (https://swissmodel.expasy.org/, accessed on 15 April 2024). Phylogenetic analysis was conducted using the Neighbor-Joining method within MEGA 6.0 software. The prediction of protein subcellular localization was carried out using Cell-PLoc 2.0 (http://www.csbio.sjtu.edu.cn/bioinf/Cell-PLoc-2/, accessed on 15 April 2024), CELLO (http://cello.life.nctu.edu.tw/, accessed on 1 November 2024), and WoLF PSORT (https://www.genscript.com/tools/wolf-psort, accessed on 1 November 2024).

### 4.4. Vector Construction

The plant expression vector was constructed using the *pEASY*^®^-Basic Seamless Cloning and Assembly Kit (TransGen Biotech, Beijing, China). The primers *LAR1*-BamH I-F and *LAR1*-XmaI-R are listed in Appendix A, and the coding sequence (CDS) of *VdLAR1* without the stop codon was inserted into the pBI121-GFP expression vector to construct a recombinant plasmid driven by the CaMV 35S promoter. The recombinant plasmid *pBI121-VdLAR1-GFP* was then transformed into *Escherichia coli* DH5α and the positive clone was selected for verification. The correctly sequenced recombinant plasmid was extracted using the Plasmid Purification Kit (TIANGEN, Beijing, China) and subsequently transferred into *Agrobacterium* GV3101 using the freeze–thaw method.

### 4.5. Subcellular Localization and Genetic Transformation 

Protoplasts were prepared from spine grape cell lines for subcellular localization analysis. The isolation of protoplast and PEG-mediated transient gene expression were conducted based on a previously described method [66], with appropriate modifications. Subcellular localization was observed using a FluoView FV1200 Confocal Laser Scanning Microscope (Olympus, Tokyo, Japan).

Agrobacterium-mediated transformation of spine grape cell lines was conducted for *VdLAR1* overexpression. Agrobacterium harboring the *LAR1* recombinant plasmid was harvested at an OD_600_ of 1.0 and resuspended in liquid MS medium containing 20 μM acetosyringone and 30 g/L sucrose until the OD_600_ was adjusted to 0.6. The cell lines were immersed in the *Agrobacterium* suspension for 8 min, then transferred onto sterilized filter paper to remove excess *Agrobacterium*-containing medium. The cell lines were co-cultured on solid MS medium supplemented with 3% sucrose, 0.6% agar, and 100 μM acetosyringone for 2 days at 25 °C in the dark. Subsequently, they were transferred to the MS selection medium containing 1.0 mg/L 2,4-dichlorophenoxyacetic acid (2,4-D), 30 g/L sucrose, 6g/L andagar, 50 μg/mL kanamycin (Kan), and 80 μg/mL terramycin (Tim). A portable fluorescent protein light source (LUYOR-3415, Shanghai, China) was used to preliminarily screen the transformed cell lines that emitted stable fluorescence. The genomic DNAs of both the wild-type (WT) and transformed cell lines were extracted using the CTAB method, and then DNA-PCR amplification was performed using vector-specific primers (Appendix A), followed by sequencing verification (TsingKe, Xiamen, China). 

### 4.6. Metabolite Determination

Wild-type (WT) and *VdLAR1*-overexpressing cell lines were cultured for 15, 25, and 35 days under different light quality conditions to evaluate the levels of proanthocyanidins, anthocyanidins, and flavonoids, as previously described [55]. The quantification of catechins and epicatechins was conducted following the instructions provided in the Enzyme-Linked Immunoassay (ELISA) Kit (Shanghai C-reagent Biotechnology, Shanghai, China).

### 4.7. Real Time qPCR Analysis

RT-qPCR analysis was conducted according to the instructions provided with the TB Green^®^
*Premix Ex Taq*™ II (TaKaRa, Dalian, China), using the primers listed in Appendix A. The qPCR reactions were performed using the LightCycler^®^ 480 Instrument II (Roche, Basel, Switzerland) with Software Version 1.5, and the crossing point (Cp) values of the amplified samples were recorded. *α-Tubulin* was chosen as the internal reference gene, and the 2^−ΔΔCt^ method was employed to calculate the expression levels of each gene.

### 4.8. Statistical Analysis

All experimental data were obtained from three independent replicates. Microsoft Excel was used to organize the data. Means were compared using SPSS 17.0 (SPSS Inc., Chicago, IL, USA) with one-way ANOVA and a two-tailed *t* test. Statistical analysis of variance (ANOVA) was performed with Duncan’s multiple range test at *p* < 0.05. Data were compared with the results of Student’s *t* test and differences were considered to be statistically significant when * *p* < 0.05 or ** *p* < 0.01. Data visualization was performed using the software GraphPad Prism 8.0, TBtools, and Photoshop CC 2019.

## 5. Conclusions

In this study, we successfully cloned the *VdLAR1* gene from spine grape, which encodes a protein with typical reductase structural features. The overexpression of *VdLAR1* promoted the biosynthesis of proanthocyanidin and flavonoid, but inhibited anthocyanin accumulation, indicating a target competitive enhancement, with a high-yield cell line obtained. Additionally, exposure to white and blue light significantly promoted the expression of *VdDFR*, *VdLDOX*, *VdLAR1*, *VdLAR2*, *VdANR*, and *VdUFGT*, leading to an increase in proanthocyanidin accumulation, with a product content of 2512.00 μg/g (FW). An efficient plant cell factory for the production of proanthocyanidins was established by utilizing a high-yield cell line and short-wavelength quality. This approach provides both theoretical and practical foundations for the regulation and development of natural proanthocyanidins, addressing issues related to high costs, safety concerns, resource wastage, and environmental damage associated with their production.

## Figures and Tables

**Figure 1 ijms-25-12087-f001:**
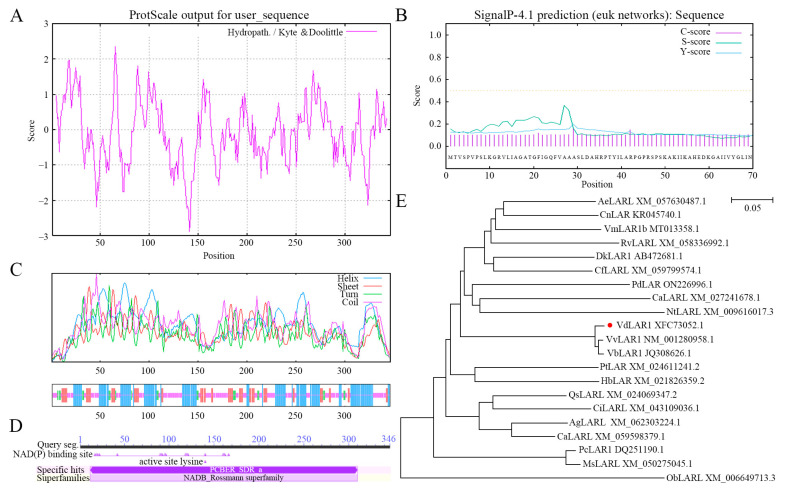
Bioinformatic analysis of VdLAR1 protein. (**A**) Hydrophilicity of VdLAR1. (**B**) Signal peptide of VdLAR1. (**C**) Prediction of protein secondary structure. (**D**) Conserved domain analysis. (**E**) The phylogenetic tree was generated using the maximum likelihood method in MEGA 6.0 software. The bar indicates a genetic distance of 0.05. The red dot indicates the VdLAR1 protein examined in this study. Plant species and GenBank accession numbers of their LAR1 proteins used for phylogenetic analysis are as follows: *Vitis davidii* (VvLAR1, XFC73052.1), *Vitis vinifera* (VvLAR1, NM_001280958.1), *Vitis bellula* (VbLAR1, JQ308626.1), *Diospyros kaki* (DkLAR1, AB472681.1), *Actinidia eriantha* (AeLARL, XM_057630487.1), *Vaccinium myrtillus* (VmLAR1b, MT013358.1), *Camellia nitidissima* (CnLAR, KR045740.1), *Populus trichocarpa* (PtLAR, XM_024611241.2), *Cornus florida* (CfLARL, XM_059799574.1), *Quercus suber* (QsLARL, XM_024069347.2), *Pyrus communis* (PcLAR1, DQ251190.1), *Alnus glutinosa* (AgLARL, XM_062303224.1), *Paeonia delavayi* (PdLAR, ON226996.1), *Malus sylvestris* (MsLARL, XM_050275045.1), *Corylus avellana* (CaLARL, XM_059598379.1), *Carya illinoinensis* (CiLARL, XM_043109036.1), *Hevea brasiliensis* (HbLAR, XM_021826359.2), *Rhododendron vialii* (RvLARL, XM_058336992.1), *Coffea arabica* (CaLARL, XM_027241678.1), *Nicotiana tomentosiformis* (NtLARL, XM_009616017.3), and *Oryza brachyantha* (ObLARL, XM_006649713.3).

**Figure 2 ijms-25-12087-f002:**
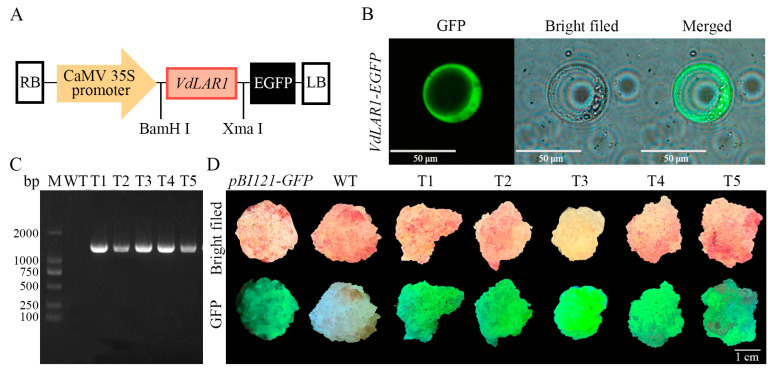
Construction of the expression vector and identification of transformed cell lines. (**A**) Schematic diagram of the expression vector. (**B**) Subcellular localization of VdLAR1 in spine grape protoplasts. Fluorescence signals were detected in the cytoplasm of the cells expressing the pBI121-*VdLAR1*-EGFP fusion gene. (**C**) PCR amplification of *VdLAR1*. (**D**) GFP fluorescence assay of GFP, non-transgenic, and transgenic spine grape cell lines.

**Figure 3 ijms-25-12087-f003:**
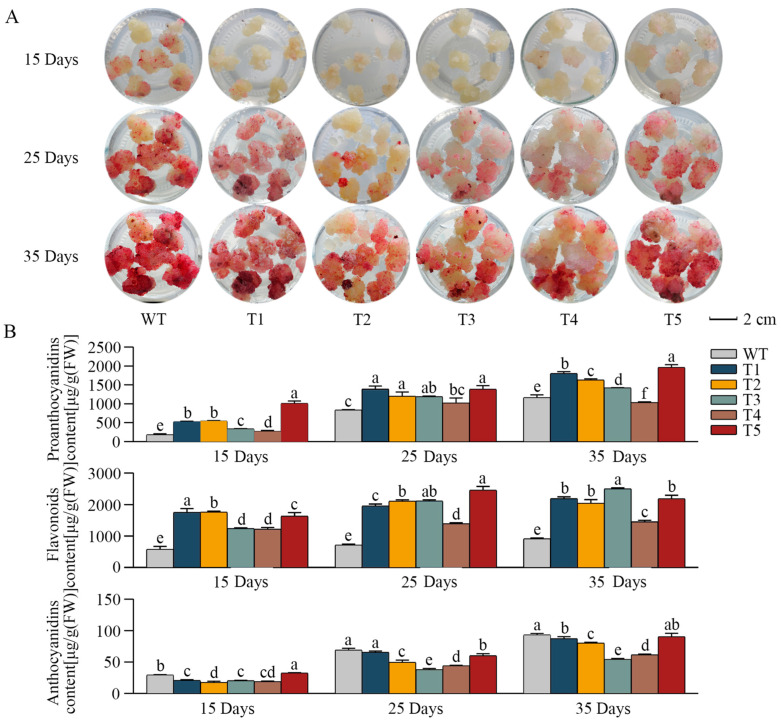
Phenotype and contents of proanthocyanidin, flavonoid, and anthocyanin in WT and transgenic cell lines. (**A**) Phenotype of WT and transgenic cell lines (T1, T2, T3, T4, and T5). (**B**) Contents of proanthocyanidin, flavonoid, and anthocyanin in WT and transgenic cell lines; different letters indicate significant differences by one-way ANOVA and Duncan’s test *p* < 0.05.

**Figure 4 ijms-25-12087-f004:**
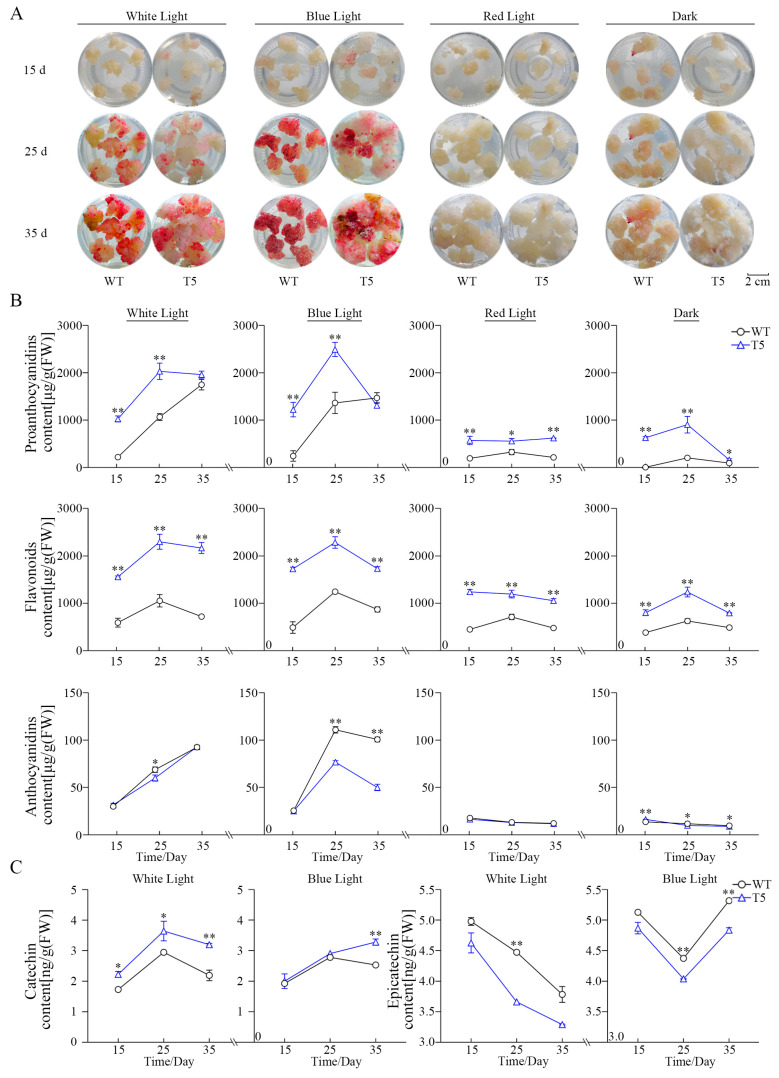
Phenotype and contents of proanthocyanidin, flavonoid, and anthocyanin in WT and T5 transgenic cell lines under different light treatments. (**A**) Phenotype of WT and T5 cell line. (**B**) Contents of proanthocyanidin, flavonoid, and anthocyanin; * for significance of *p* < 0.05 and ** for significance of *p* < 0.01. (**C**) Contents of catechin and epicatechin; * for significance of *p* < 0.05 and ** for significance of *p* < 0.01.

**Figure 5 ijms-25-12087-f005:**
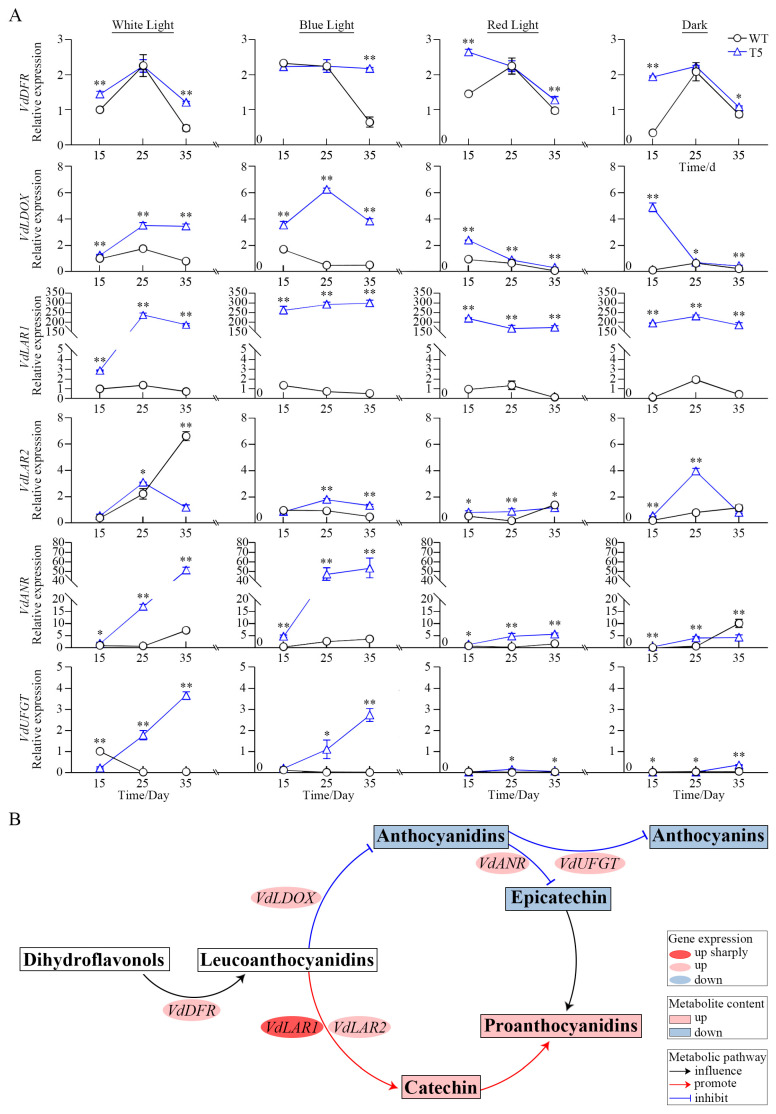
Relative gene expression involved in the proanthocyanidin pathway. (**A**) The gene expression model in the *VdLAR1*-transformed cell line under different light treatments; * for significance of *p* < 0.05 and ** for significance of *p* < 0.01. (**B**) Gene expression and metabolite accumulation related to proanthocyanidin and anthocyanin biosynthesis. Overexpression of *VdLAR1* promoted catechin biosynthesis, enhanced proanthocyanidin levels, and inhibited anthocyanin production.

## Data Availability

The VdLAR1 sequence in this study are openly available in GenBank at https://www.ncbi.nlm.nih.gov/nuccore/PQ185972.1/ (accessed on 15 April 2024), reference number PQ185972.

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
