# Peer review of "Overexpression of LAR1 Suppresses Anthocyanin Biosynthesis by Enhancing Catechin Competition Leading to Promotion of Proanthocyanidin Pathway in Spine Grape (Vitis davidii) Cells"

_ijms, 2024, doi:10.3390/ijms252212087_

Round 1
Reviewer 1 Report
Comments and Suggestions for Authors
Dear Authors,
Reviewer comments ijms-3260132
The manuscript entitled „Overexpression of LAR1 suppresses anthocyanin biosynthesis by enhancing catechin competition leading to promotion of proanthocyanidin pathway in spine grape (Vitis Davidii) cells“ represents a useful study aimed at a comparison of WT and LAR1 overexpressing T5 spine grape line responses to the four light quality regimes (white, red, blue light and darkness) with respect to anthocyanin and catechins content. The results of the study indicate that LAR1 overxpression leads to an enhancement of proanthocyanidin pathway while decreasd anthocyanin accumulation. I can recommend the manuscript for publication in IJMS.
I have only a few minor comments on the present manuscript which are provdied below:
1/ In Materials and methods, part 4.1., the wavelengths ranges have to be provided for the blue and red light treatments. How were the differential spectra ensured by a single type of a light-emitting diode??
2/ In Materials and methods, line 398, the date of access has to eb added to a reference grapevine genome sequence.
3/ In Materials and methods, line 400, a reference is missing in the statement that „specific primers designed using Primer Premier 5.0 to amplify the full-length cDNA are listed in ??? – most probably Table S1 which is referred at line 415.
4/ In Materials and methods, line 448, correct the typing error in „Enzyme-linked Immunoassay (ELISA) kit“.
5/ In Results, Figure 1E, the bootstrap values in the phylogenetic tree mentioned in the legend are missing in the tree; however, I think that the scale bar providing teh information on genetic distance is sufficient.
6/ In Results, Figure 4B,C and Figure 5A, the significant differences between WT and T5 line have to be indicated by asterisks based on the Student T-test designed for a comaprison of two varieties.
7/ Formal comments on the text related to English language and style:
Abstract, line 31: Add a noun and correct the verb form in the statement „This finding provides compelling evidence that LAR1 is a cucial gene for catechin biosynthesis.“
Abstract, line 32. Correct the verb form „establishs“ to „establishes“ in the statement: „This research establishes both theoretical and practical foundations for the rgulation and development…“
Introduction, line 43: Remove the word „in“ in the statement: „As a result, they are crucial in mechanisms associated with anti-inflammation processes,…“
Results, line 162: Modify the word form „decrease levels“ to „decreased levels“.
Results, line 188: Add „an“ and „a“ preceding the words „increase“ and „decrease“, respectively, in the statement: „…and the accumulation exhibited a pattern of an increase followed by a decrease….“
Results, line 200: Add „a“ preceding the word „decrease“ in the statement „…followed by a decrease“.
Results, line 218: Add „an“ preceding the words „initial decrease“, i.e., „an initial decrease“.
Results, line 244. Add „a“ preceding the words „significant downregulation“, i.e., „a significant downregulation“.
Discussion, line 335. Add the word „result“ following „this“ in the statement: „This result provides robust evidence that LAR1 is a crucial gene for catechin biosynthesis.“
Discussion, line 350. Replace the word „heightened“ with „enhanced“ in the statement „and enhanced flavonoid accumulation.“
Materials and methods, line 435. Add a comma following the word „Subsequently“, i.e., „Subsequently, they were transferred to…“
Materials and methods, line 448: Correct the typing error in the term „Enzyme-linked Immunoassay (ELISA) kit“.
Final recommendation: Accept after a minor revision.
Author Response
Dear reviewer,
we are deeply grateful for your meticulous work and thoughtful suggestions that have helped us to significantly improve this manuscript. In terms of the comments, we have answered all questions one by one (please see the below in blue color). Based on these comments and suggestions, we have made careful modifications (please see the red text in the revised manuscript).
Comments 1: In Materials and methods, part 4.1., the wavelengths ranges have to be provided for the blue and red light treatments. How were the differential spectra ensured by a single type of a light-emitting diode??
Response 1: Dear reviewer, we appreciate your insightful feedback regarding this matter. We fully concur with your concern and have provided the wavelength ranges for the light treatments, revising the relevant section of the manuscript as follows (line 421-423): “Four types of light quality were employed: white light (mix light), blue light (400-500 nm), red light (620-700 nm) and darkness, with different technical parameters for light quality as previously described [55].” In addition, the light intensity range was controlled about 2000 Lux by adjusting the number of LED lamps. A HiPoint spectrometer (HiPoint Biotechnology Co., Ltd., Xiamen, China) was used to determine the technical parameters such as light intensity, wavelength peak, and photosynthetic photon flux density (PPFD), which were mentioned in the paper previously published by this laboratory [50], as shown in the figure below.

References
- Lai, C. C.; Pan, H.; Zhang, J.; Wang, Q.; Que, Q. X.; Pan, R.; Lai, Z. X.; Lai, G. T. Light quality modulates growth, triggers differential accumulation of phenolic compounds, and changes the total antioxidant capacity in the red callus of Vitis davidii. Journal of agricultural and food chemistry 2022,70, (41), 13264-13278.
Comments 2: In Materials and methods, line 398, the date of access has to be added to a reference grapevine genome sequence.
Response 2: Dear reviewer, we have added the date of access and revised the sentence as follows (line 437-438): Based on the reference grapevine genome sequence (Genoscope 12X v2.1) in March 2023.
Comments 3: In Materials and methods, line 400, a reference is missing in the statement that „specific primers designed using Primer Premier 5.0 to amplify the full-length cDNA are listed in ??? -most probably Table S1 which is referred at line 415.
Response 3: Dear reviewer, thank you for your reminder. We have added “Table S1” at line 439.
Comments 4: In Materials and methods, line 448, correct the typing error in “Enzyme-linked Immunoassay (ELISA) kit”.
Response 4: Dear reviewer, we have revised “Enzyme-linked Mmmunoassay (ELISA) kit” to “Enzyme-linked Immunoassay (ELISA) kit” at line 492.
Comments 5: In Results, Figure 1E, the bootstrap values in the phylogenetic tree mentioned in the legend are missing in the tree; however, I think that the scale bar providing the information on genetic distance is sufficient.
Response 5: Dear reviewer, thank you for pointing this out. The bootstrap values are not indicated in the graph, and we only provided the scale bar on genetic distance. In accordance with your suggestion, we have removed the sentence “Numbers at each interior branch indicate the bootstrap values of 1000 replicates.” in the legend at line 118 while retaining the scale bar for genetic distance.
Comments 6: In Results, Figure 4B,C and Figure 5A, the significant differences between WT and T5 line have to be indicated by asterisks based on the Student T-test designed for a comaprison of two varieties.
Response 6: We appreciate your valuable suggestion. We have indicated the significant differences between WT and T5 lines and updated Figure 4 and 5 in the revised manuscript. Additionally, we have included a description of the Student T-test methodology as follows (line 503-507): Means were compared using SPSS 17.0 (SPSS Inc., Chicago, USA) with the one-way ANOVA and two-tailed t test. Statistical analysis of variance (ANOVA) was performed with Duncan’s multiple range test at p < 0.05. Data were compared with the student’s t test and differences were considered to best atistically significant when *p < 0.05 or **p < 0.01.
Comments 7: Formal comments on the text related to English language and style:
Abstract, line 31: Add a noun and correct the verb form in the statement“This finding provides compelling evidence that LAR1 is a cucial gene for catechin biosynthesis.”
Abstract, line 32: Correct the verb form “establishs” to “establishes” in the statement: “This research establishes both theoretical and practical foundations for the rgulation and development…”
Introduction, line 43: Remove the word “in” in the statement: “As a result, they are crucial in mechanisms associated with anti-inflammation processes,…”
Results, line 162: Modify the word form “decrease levels” to “decreased levels”.
Results, line 188: Add “an” and “a” preceding the words “increase” and “decrease”, respectively, in the statement: “…and the accumulation exhibited a pattern of an increase followed by a decrease….”
Results, line 20,0: Add “a” preceding the word “decrease” in the statement “…followed by a decrease”.
Results, line 218: Add “an” preceding the words “initial decrease”, i.e., “an initial decrease”.
Results, line 244. Add “a” preceding the words “significant downregulation”, i.e., “a significant downregulation”.
Discussion, line 335. Add the word “result” following “this” in the statement: “This result provides robust evidence that LAR1 is a crucial gene for catechin biosynthesis.”
Discussion, line 350. Replace the word “heightened” with “enhanced” in the statement “and enhanced flavonoid accumulation.”
Materials and methods, line 435. Add a comma following the word “Subsequently”, i.e., “Subsequently, they were transferred to…”
Materials and methods, line 448: Correct the typing error in the term “Enzyme-linked Immunoassay (ELISA) kit”.
Response 7: Dear reviewer, we have corrected all errors in revised manuscript at line 32, 33, 45, 168, 195, 208, 226, 257, 356, 387, 472, and 492.
Reviewer 2 Report
Comments and Suggestions for Authors
Comments
The manuscript " Overexpression of LAR1 Suppresses Anthocyanin Biosynthesis by Enhancing Catechin Competition Leading to Promotion of Proanthocyanidin Pathway in Spine Grape (Vitis davidii) Cells " represents good and very interesting work. However, the authors need to make some modifications before considering the manuscript for publication.
1. The authors need to make identification and quantification of individual anthocyanins and non-anthocyanin phenolics (individual Proanthocyanidins) using HPLC-MS/MS to confirm. ELISA assay is not enough.
Author Response
Dear reviewer,
we are deeply grateful for your meticulous work and thoughtful suggestions that have helped us to significantly improve this manuscript. In terms of the comments, we have answered all questions one by one (please see the below in blue color).
Comment 1: The authors need to make identification and quantification of individual anthocyanins and non-anthocyanin phenolics (individual Proanthocyanidins) using HPLC-MS/MS to confirm. ELISA assay is not enough.
Response 1: Dear reviewer, we sincerely appreciate your insightful reminder regarding the impact of overexpression on anthocyanin monomers and proanthocyanidin monomers, we fully understand your concern. In this study, to investigate the influence of overexpression on secondary metabolites content, we initially quantified the total contents of proanthocyanidins, anthocyanidins and flavonoids in the cell line. Because LAR1 is a key gene that promotes proanthocyanidin synthesis in plant cells, its direct downstream product is catechin, and the synthesis of catechin directly affects the accumulation of total proanthocyanidin. Therefore, we concentrated our efforts on examining the effects of VdLAR1 on both catechin levels and overall proanthocyanidin content. Consequently, after observing a significant increase in total proanthocyanidin levels within overexpressed cell lines, we also assessed and analyzed catechin along with its competitive metabolite, epicatechin. Currently, ELISA kits are widely used, and this journal has also published papers utilizing this detection method [1]. Furthermore, numerous studies have successfully applied ELISA kits to quantify various plant metabolites, including anthocyanins, chlorophylls, and melatonin, among others [2-4]. By studying the effects of VdLAR1 overexpression on catechin pathway-related genes and products, we aim to establish a proanthocyanidin-rich cell line and provide a theoretical and practical basis for the regulation and development of natural proanthocyanidins. In response to your constructive feedback, we will incorporate HPLC-MS/MS experiments in our future similar studies to further substantiate our research findings. We hope that our research content and methodologies will gain your recognition.
References
- Xu, Y.; Wang, R.; Ma, Y.; Li, M.; Bai, M.; Wei, G.; Wang, J.; Feng, L. Metabolite and Transcriptome Profiling Analysis Provides New Insights into the Distinctive Effects of Exogenous Melatonin on Flavonoids Biosynthesis in Rosa rugosa. Int. J. Mol. Sci 2024, 25, 9248.
- Chen, L.; Lu, B.; Liu, L. Melatonin promotes seed germination under salt stress by regulating ABA and GA3 in cotton (Gossypium hirsutum L.). Plant Physiology and Biochemistry 2021, 2, 162.
- Li, R. Q.; Jiang, M.; Huang, J. Z.; Møller, I. M.; Shu, Q. Y. Mutations of the Genomes Uncoupled 4 Gene Cause ROS Accumulation and Repress Expression of Peroxidase Genes in Rice. Frontiers in plant science 2021, 12, 682453.
- Liu, P.; Gang, H.; Liu, H.; Qin, D.; Zhang, Y.; Huo, J. Regulation of Anthocyanin Accumulation by a Transcription Factor LcTT8 From Lonicera caerulea L. Plant Molecular Biology Reporter 2020, 39(1), 1-12.
Reviewer 3 Report
Comments and Suggestions for Authors
In their manuscript titled "Overexpression of LAR1 Suppresses Anthocyanin Biosynthesis by Enhancing Catechin Competition Leading to Promotion of Proanthocyanidin Pathway in Spine Grape (Vitis Davidii) Cells," Lin et al. explore the role of VdLAR1 in regulating proanthocyanidin and flavonoid biosynthesis in spine grape. They began by conducting a bioinformatic analysis of the VdLAR1 protein, followed by testing its subcellular localization, which revealed its presence in the cytoplasm. Callus overexpressing VdLAR1 exhibited increased levels of proanthocyanidin and flavonoids, particularly under short-wavelength light treatment, such as blue light. The authors assert that blue light significantly enhanced the expression of VdDFR, VdLDOX, VdLAR1, VdLAR2, VdANR, and VdUFGT, leading to a notable accumulation of proanthocyanidin. The primary goal of this study is to offer new potential strategies for enhancing proanthocyanidin production. While the title and abstract align well with the content, and the experiments were generally well-executed, some modifications to the figures and text would improve clarity and readability. Additionally, some figures are unclear, and the conclusions presented may not be fully supported by the experimental design and data. The experimental design requires further attention for the conclusions to be convincingly established.
Major Concerns:
The authors have presented a high-quality manuscript that is logically structured and enjoyable to read. However, a significant concern is that the data presented may not be sufficient to support the major conclusions. The RT-qPCR results for proanthocyanidin biosynthesis-related genes and the levels of proanthocyanidin, flavonoid, and anthocyanin in response to different light treatments were tested only in VdLAR-OE callus. Testing VdLAR1-RNAi or CRISPR knockout mutants would provide stronger support for the conclusions. Additionally, for practical applications, the high cost of callus induction and low transformation efficiency are significant obstacles. It would be valuable to compare the proanthocyanidin production efficiency of the VdLAR-OE cell lines to that of the skin and seeds of spine grape.
Minor Concerns:
- In Fig.1, please ensure consistent fonts and line widths across all panels.
- In Fig.1B, the prediction shown is not meaningful and should be removed, as Fig. 2B shows that VdLAR1 is localized in the cytoplasm. More evidence is needed to support any claim of transmembrane or cell membrane localization.
- In line 129, the construct pBI121-VdLAR1-EGFP should be italicized.
- "Positive cell lines" should be referred to as "positive calluses."
- In Fig. 2C, please clarify why VdLAR1 could not be amplified from wild-type DNA. Also, the PCR fragments appear to exceed 1041 bp; please label the DNA ladder marker sizes and explain the primer design.
- In Fig. 2D, please label the raw GFP fluorescence and bright-field images.
- A citation is missing for Fig. 3B in the text.
- In line 153-154, the statement that "Metabolite determination revealed significantly higher proanthocyanidin content in transformed cell lines compared to WT" is inaccurate, as the T4 cell line shows lower proanthocyanidin levels than WT at 35 days.
- In line 156-158, the authors claim that "T5 demonstrated the highest accumulation of proanthocyanidin, with an increase ranging from 1.7 to 5.5-fold and a peak concentration of 1960.89 μg/g (FW) at the late culture stage (35 d)." However, the data in Fig. 3B do not support this 1.7 to 5.5-fold increase at 35 days.
- In line 159, please specify the time points corresponding to the reported fold increases ("1.5 to 3.54-fold").
- In line 165, regarding anthocyanins in Fig. 3B, the T3 cell line shows lower levels at two of the three time points. Please explain why T4 is emphasized here.
- In line 231-233, the statement that "RT-qPCR results revealed significantly higher expression of VdDFR, VdLDOX, VdLAR1, VdLAR2, VdANR, and VdUFGT in the T5 cell line compared to WT" is not fully supported, as VdLAR2 expression at 35 days under white light is significantly lower than in WT, and no significant difference is observed under red light or dark conditions at 35 days.
- In line 246-247, the statement that "VdANR and VdUFGT exhibited nearly consistent expression patterns" is unclear and not fully supported by the data. Please revise.
- In line 429, LAR1 should be VdLAR1.
- In line 419, the rationale for transforming E. coli plasmids instead of pBI121-35S::VdLAR1 into GV3101 is unclear. Please clarify.
Author Response
Dear reviewer,
we are deeply grateful for your meticulous work and thoughtful suggestions that have helped us to significantly improve this manuscript. In terms of the comments, we have answered all questions one by one (please see the below in blue color). Based on these comments and suggestions, we have made careful modifications (please see the red text in the revised manuscript).
Major Concerns: The authors have presented a high-quality manuscript that is logically structured and enjoyable to read. (1) However, a significant concern is that the data presented may not be sufficient to support the major conclusions. The RT-qPCR results for proanthocyanidin biosynthesis-related genes and the levels of proanthocyanidin, flavonoid, and anthocyanin in response to different light treatments were tested only in VdLAR-OE callus. Testing VdLAR1-RNAi or CRISPR knockout mutants would provide stronger support for the conclusions. (2) Additionally, for practical applications, the high cost of callus induction and low transformation efficiency are significant obstacles. (3) It would be valuable to compare the proanthocyanidin production efficiency of the VdLAR-OE cell lines to that of the skin and seeds of spine grape.
Response: Dear reviewer, thank you very much for your constructive comments.
(1) Firstly, we acknowledge your concern regarding the limited experimental data obtained solely from the VdLAR-OE cell line, which may not be sufficient. Currently, the primary objective of this study is to establish high-yield proanthocyanidin cell line through the overexpression of VdLAR1 and light treatment, thereby providing technical support for large-scale industrial production of proanthocyanidins. Furthermore, it is valuable to further validate our findings using VdLAR1-RNAi or CRISPR knockout mutants as experimental materials, so we will actively conduct related experiments in the future.
(2) Secondly, we appreciate your concerns regarding the high costs of callus induction and the low transformation efficiency, which present significant challenges for practical applications. However, we have made every effort to overcome these obstacles, and successfully obtained various grape calli and constructed feasible genetic transformation methods, particularly for spine grape. Recently, we reported our research involved on callus induction [1] and genetic transformation [2]. Additionally, we have developed numerous transgenic cell lines, including VdANR, VdF3'5'H, VdUFGT, VdSTS. Therefore, based on the established methods for callus induction and genetic transformation, we will continue to focus on developing high-yielding secondary metabolite cells through genetic engineering, thus providing technical support for the production and application of natural bioactive compounds.
References
- Lai, C. C.; Pan, H.; Zhang, J.; Wang, Q.; Que, Q. X.; Pan, R.; Lai, Z. X.; Lai, G. T. Light quality modulates growth, triggers differential accumulation of phenolic compounds, and changes the total antioxidant capacity in the red callus of Vitis davidii. Journal of agricultural and food chemistry 2022, 70, (41), 13264-13278.
- Lai, T.; Fu, P. N.; He, L. Y.; Che, J. M.; Wang, Q.; Lai, P. F.; Lai, C. C. CRISPR/Cas9 mediated CHS2 mutation provides a new insight into resveratrol biosynthesis by causing a metabolic pathway shift from flavonoids to stilbenoids in Vitis davidii cells, Horticulture Research 2024, uhae, 268.
(3) Finally, we particularly agree with your suggestion to compare the proanthocyanidin production efficiency of the VdLAR-OE cell lines to that of the skin and seeds of spine grape. In response, we compared the production efficiency of proanthocyanidins in VdLAR-OE cell lines, skin and seeds of spine grape in the discussion section (line 357-372):
As a result, the findings demonstrated that the overexpression of VdLAR1 significantly enhanced the levels of both monomeric anthocyanins and total proanthocyanidins in cell lines. Previous studies have indicated that proanthocyanidins are abundant in spine grape skin and seeds, with concentrations approximately 50,000 μg/g (DW) and 10,000 μg/g (DW), respectively [47, 48]. Notably, the proanthocyanidin content in overexpressing VdLAR1 cell line can reach up to 2,000 μg/g (FW), and due to a water content exceeding 95%, the dry weight concentration of proanthocyanidins in these samples surpasses 40,000 μg/g (DW), which is significantly higher than that found in grape seeds and comparable to that present in skins [49]. Furthermore, the high costs associated with removal of skins and seeds from spine grapes, coupled with seasonal growth limitations, adversely affect the production efficiency of proanthocyanidins derived from these sources [50]. In contrast, callus cultures remain unaffected by external environmental conditions, allowing for year-round continuous production [51]. Therefore, transforming callus for efficient proanthocyanidin production through overexpression presents promising application prospects.
References
- Tan, S.; Zhu, R.W.; Yin, R.L.; Li, G.F.; Li, L.; Zhu, M. Analysis on Nutritional Component of Vitis davidii Foex. Guangzhou Chemical Industry 2017, 21, 124-126.
- Ma, H.; Hou, A.; Tang, J.; Zhong, A.; Li, K.; Xiao, Y.; Li, Z. Antioxidant activity of Vitis davidii Foex seed and its effects on gut microbiota during colonic fermentation after in vitro simulated digestion. Foods 2022, 11, 2615.
- Lai, C.; Zhang, J.; Lai, G.; He, L.; Xu, H.; Li, S.; Che, J.; Wang, Q.; Guan, X.; Huang, J.; Lai, P.; Chen, G. Targeted regulation of 5-aminolevulinic acid enhances flavonoids, anthocyanins and proanthocyanidins accumulation in Vitis davidii callus. BMC plant biology 2024, 24, (1), 944.
- Cheng, J.; Xiang, J.; Wei, L.; Zheng, T.; Wu, J. Metabolomic profiling and assessment of phenolic compounds derived from Vitis davidii Foex cane and stem extracts. International journal of molecular sciences 2022, 23, (23), 14873.
- Efferth, T. Biotechnology applications of plant callus cultures. Engineering 2019, 5, (1), 50-59.
Comments 1: In Fig.1, please ensure consistent fonts and line widths across all panels.
Response 1: Dear reviewer, thank you for your valuable suggestion and we have updated Figure 1 in the revised manuscript.
Comments 2: In Fig.1B, the prediction shown is not meaningful and should be removed, as Fig. 2B shows that VdLAR1 is localized in the cytoplasm. More evidence is needed to support any claim of transmembrane or cell membrane localization.
Response 2: Dear reviewer, thank you for pointing this out. we have removed the prediction of transmembrane structure and updated Figure 1 in revised manuscript.In addition, we have added more subcellular localisation websites in the methods section as follows (line 452-454): The prediction of protein subcellular localization was carried out using Cell-PLoc 2.0 ( http://www.csbio.sjtu.edu.cn/bioinf/Cell-PLoc-2/) , CELLO (http://cello.life.nctu.edu.tw/), and WoLF PSORT (https://www.genscript.com/tools/wolf-psort).
Furthermore, we have provided a more detailed explanation in the discussion section as follows (line 305-311): Regarding the subcellular localization of VdLAR1, all three prediction tools (Cell-PLoc 2.0 , CELLO, and WoLF PSORT) indicated that VdLAR1 was localized in the cytoplasm. This finding was corroborated by the results of transient transformation into protoplasts conducted in this study. Moreover, the integration of bioinformatics analysis and the examination of LAR subcellular localization across different species further confirmed the presence of LAR proteins in the cytoplasm [33-35].
References
- Li, H.; Tian, J.; Yao, Y. Y.; Zhang, J.; Song, T. T.; Li, K. T.; Yao, Y. C. Identification of leucoanthocyanidin reductase and anthocyanidin reductase genes involved in proanthocyanidin biosynthesis in Malus crabapple plants. Plant physiology and biochemistry : PPB 2019, 139, 141–151.
- Xin, Y.; Meng, S.; Ma, B.; He, W.; He, N. Mulberry genes MnANR and MnLAR confer transgenic plants with resistance to Botrytis cinerea. Plant science : an international journal of experimental plant biology 2020, 296, 110473.
- Zhu, L. J.; Wang, Z.; Wang, Z. Y.; Zhang, Y. T.; Huang, J.; Xing, C. B.Cloning and expression of LAR gene and its correlation with phloridzin content in Lithocarpus polystachyus. Chinese Traditional and Herbal Drugs 2020, 47, (7), 953–961.
Comments 3: In line 129, the construct pBI121-VdLAR1-EGFP should be italicized.
Response 3: Dear reviewer, we have revised “pBI121-VdLAR1-EGFP” to “pBI121-VdLAR1-EGFP” in line 132.
Comments 4: “Positive cell lines” should be referred to as “positive calluses.”
Response 4: Dear reviewer, we have revised “Positive cell lines” to “positive calli” in line 143.
Comments 5: In Fig. 2C, please clarify why VdLAR1 could not be amplified from wild-type DNA. Also, the PCR fragments appear to exceed 1041 bp; please label the DNA ladder marker sizes and explain the primer design.
Response 5: Dear reviewer, we have included a description explaining why VdLAR1 could not be amplified from wild-type DNA as follows (line 140-143): “The DNA-PCR amplification confirmed that the electrophoretic bands of all the transformed cell lines matched the expected results, while the WT exhibited no electrophoretic band due to the absence of vector sequences that harbouring the targeted LAR1 gene”. Additionally, we have labeled the DNA ladder marker sizes in Figure 2C and updated this figure in revised manuscript. Furthermore, the PCR fragment length exceeding 1041 bp because the DNA validation primer was located on the pBI121 vector, and the obtained target fragment contained part of the vector sequence. In responds, we added a description of DNA validation primers as follows (line 484-485): the genomic DNAs of both the wild-type (WT) and transformed cell lines were extracted using the CTAB method, and then DNA-PCR amplification was performed using vector-specific primers (Table S1), followed by sequencing verification (TsingKe, Xiamen, China). The sequencing results of DNA-PCR amplification are as follows:
(1-22 bp and 1204-1225 bp: vector-specific primers; 23-70 bp and 1109-1203 bp: vector sequences; 71-1108 bp: the CDS of VdLAR1 without stop codon)
TCCTTCGCAAGACCCTTCCTCTATATAAGGAAGTTCATTTCATTTGGAGAGAACACGGGGGACTCTAGAGATGACTGTTTCTCCGGTTCCTTCGCTCAAGGGTCGTGTCCTCATTGCCGGAGCAACCGGTTTCATTGGTCAGTTCGTGGCCGCAGCAAGCCTTGATGCCCATCGACCCACCTACATTCTCGCACGTCCAGGCCCCAGGAGTCCTTCTAAGGCCAAGATCATCAAGGCCCACGAGGACAAAGGCGCCATCATCGTATACGGGTTGATAAACGAGCAGGAGTCTATGGAGAAGATACTAAAAGAACATGAGATAGACATAGTAGTATCAACCGTGGGCGGAGAGAGCATATTGGATCAAATCGCCCTAGTGAAAGCCATGAAGGCTGTTGGAACCATTAAGAGATTTTTGCCGTCTGAATTCGGGCACGATGTGAACAGAGCTGATCCAGTTGAGCCAGGGCTCAACATGTACAGAGAGAAGCGTAGGGTCCGACAATTTGTGGAGGAATCGGGCATACCCTTCACTTATATCTGCTGCAACTCAATTGCTTCTTGGCCATACTACAATAACATTCACCCTTCTGAGGTTCTTCCTCCAACGGATTTCTTCCAGATTTACGGTGATGGCAATGTCAAAGCTTACTTTGTTGCAGGCACAGACATCGGAAAATTCACGATGAAAACAGTGGACGATGTCCGAACACTGAACAAATCAGTGCATTTCCGGCCATCTTGCAATTGTCTCAACATAAATGAACTCGCATCTGTGTGGGAAAAGAAGATTGGGAGGACACTTCCCAGAGTAACCGTCACTGAAGATGATCTACTAGCTGCAGCCGGAGAAAACATCATCCCACAGAGTGTGGTAGCGGCGTTCACGCACGACATTTTCATAAAGGGGTGTCAGGTGAATTTCTCTATTGATGGCCCGGAGGACGTGGAAGTGACCACCCTCTACCCTGAGGATTCTTTCAGGACGGTGGAGGAATGCTTCGGCGAATACATCGTGAAGATAGAGGAAAAGCAGCCGACCGCCGATTCTGCTATTGCCAACACCGGTCCTGTGGTTGGGATGCGGCAAGTCACTGCAACCTGCGCTGTACCGGTCGCCACCATGGTACCGGTCGCCACCATGGTGAGCAAGGGCGAGGAGCTGTTCACCGGGGTGGTGCCCATCCTGGTCGAGCTGGACGGCGACGTAAACGGCCACAAGTTC
Comments 6: In Fig. 2D, please label the raw GFP fluorescence and bright-field images.
Response 6: Dear reviewer, thank you for your valuable suggestion, we have labeled the raw GFP fluorescence and bright-field images and updated Figure 2D in revised manuscript.
Comments 7: A citation is missing for Fig. 3B in the text.
Response 7: Dear reviewer, we have add citations for Fig. 3B in line 160, 165, and 169.
Comments 8: In line 153-154, the statement that "Metabolite determination revealed significantly higher proanthocyanidin content in transformed cell lines compared to WT" is inaccurate, as the T4 cell line shows lower proanthocyanidin levels than WT at 35 days.
Response 8: Dear reviewer, thank you for your reminder. We have revised the description as follows (line 159-160): Metabolite determination revealed a significantly higher proanthocyanidin content in transgenic cell lines compared to WT, except for T4 at the late culture stage (35 d).
Comments 9: In line 156-158, the authors claim that "T5 demonstrated the highest accumulation of proanthocyanidin, with an increase ranging from 1.7 to 5.5-fold and a peak concentration of 1960.89 μg/g (FW) at the late culture stage (35 d)." However, the data in Fig. 3B do not support this 1.7 to 5.5-fold increase at 35 days.
Response 9: We appreciate your observation and concur with this comment. The increase of 1.7 to 5.5-fold described in the article indicates that, at the three cultivation stages (15 d, 25 d, and 35 d), our previous expression may have led to some misunderstanding. Therefore, we have revised the description as follows (line 162-164): Among these transformed cell lines, T5 demonstrated the highest accumulation of proanthocyanidin, which reached the peak concentration of 1960.89 μg/g (FW) at the late culture stage (35 d), an increase of 1.7-fold.
Comments 10: In line 159, please specify the time points corresponding to the reported fold increases ("1.5 to 3.54-fold").
Response 10: Dear reviewer, 1.5-fold and 3.54-fold were at 25 d and 35 d respectively, so we have revised the sentence as follows (line 165): The flavonoid levels showed a significant enhancement in the transformed cell lines, with a maximum increase of 3.54-fold at the late culture stage (35 d).
Comments 11: In line 165, regarding anthocyanins in Fig. 3B, the T3 cell line shows lower levels at two of the three time points. Please explain why T4 is emphasized here.
Response 11: Dear reviewer, we appreciate your reminder and apologize for the oversight in writing T3 instead of T4. And we have corrected this error and revised the sentence as follows (line 171): The T5 cell line also demonstrated relatively high levels at 90.52 μg/g (FW), whereas the T3 cell line exhibited only 40.9% of the WT content, measuring 54.09 μg/g (FW).
Comments 12: In line 231-233, the statement that "RT-qPCR results revealed significantly higher expression of VdDFR, VdLDOX, VdLAR1, VdLAR2, VdANR, and VdUFGT in the T5 cell line compared to WT" is not fully supported, as VdLAR2 expression at 35 days under white light is significantly lower than in WT, and no significant difference is observed under red light or dark conditions at 35 days.
Response 12: Dear reviewer, we have revised the sentence as follows (line 241-246): RT-qPCR results revealed that the expression levels of VdDFR, VdLDOX, VdLAR1, VdLAR2, VdANR, and VdUFGT genes were significantly higher in the T5 cell line compared to WT. However, exceptions were observed for VdLAR2 under white light, blue light, and darkness at the late culture stage (35 d), as well as for VdANR under darkness at this stage and for VdUFGT under white light at the early culture stage (15 d) (Figure 5A).
Comments 13: In line 246-247, the statement that "VdANR and VdUFGT exhibited nearly consistent expression patterns" is unclear and not fully supported by the data. Please revise.
Response 13: Dear reviewer, thank you for pointing this out. We have revised the sentence as follows (line 259-265): The expression levels of VdANR and VdUFGT increased under both white and blue light, while remaining at low expression level under red light and in darkness. VdANR exhibited an increased expression pattern at all three cultivation stages (15 d, 25 d, and 35 d) compared to WT. While VdUFGT showed a similar expression pattern, except at the early cultivation stage (15 d), where it displayed a low level of expression under white light.
Comments 14: In line 429, LAR1 should be VdLAR1.
Response 14: Dear reviewer, we have revised “LAR1” to “VdLAR1” in line 472.
Comments 15: In line 419, the rationale for transforming E. coli plasmids instead of pBI121-35S::VdLAR1 into GV3101 is unclear. Please clarify.
Response 15: Dear reviewer, thank you for pointing this out. The E. coli plasmid in this sentence refers to the recombinant plasmid pBI121-VdLAR1-EGFP, so we have revise0d the sentence as follows (line 460-464): The recombinant plasmid pBI121-VdLAR1-GFP was then transformed into Escherichia coli DH5α and the positive clone was selected for verification. The correctly sequenced recombinant plasmid was extracted using the Plasmid Purification Kit (TIANGEN, Beijing, China) and subsequently transferred into Agrobacterium GV3101 using the freeze-thaw method.